# Canadian natural science graduate stipends lie below the poverty line

**Andrew J. Fraass**[1,2,3]*⊙‡, **Thomas J. Bailey**[4,5]⊙‡, **Kayona Karunakumar**[4],
**Andrea E. Wishart**[5,6]

**1** School of Earth and Ocean Sciences University of Victoria, British Columbia, Victoria, Canada, **2** School of Earth Sciences, University of Bristol, Bristol, England, **3** Invertebrate Paleontology, The Academy of Natural Sciences of Drexel University, Philadelphia, Pennsylvania, United States of America, **4** Department of Physics, University of Ottawa, Ottawa, Ontario, Canada, **5** Support Our Science, Ottawa, Ontario, Canada, **6** Department of Biology, University of Saskatchewan, Saskatoon, Saskatchewan, Canada

⊙ These authors contributed equally to this work.
‡ These authors are joint senior authors on this work.
* andyfraass@uvic.ca

## Abstract

Despite the critical role of graduate students in the Canadian research ecosystem, students report high levels of financial stress. As a case study, we collected graduate minimum stipends and tuition data from all university graduate programs in Canada in Ecological Sciences/Biology and Physics, along with cost of living measures for the cities in which they reside. These data are heterogeneous, complex, and in many cases simply not publicly available, making it challenging for potential graduate students to understand what support they should expect. We find Canadian minimum stipends are at values almost exclusively below the poverty threshold. Only two of 140 degree programs offered stipends which meet cost of living measures after subtracting tuition and fees. For graduate programs which offered a minimum guaranteed stipend, the average minimum domestic stipend is short ~Can$9,584 (international ~Can$16,953) of the poverty threshold after accounting for payment of tuition and fees. On average, approximately 33% of a minimum stipend is returned to the university in tuition and fees by a domestic Canadian student and 76% (59% median) by an international student, though there are important caveats with the international student comparison. While international comparison is difficult, the highest Canadian minimum stipend found is roughly equivalent or lower than the lowest stipend within the largest dataset of United States of America (US) Biology stipends, and lower than the United Kingdom (UK) stipend. University endowment correlates with minimum stipend amount but intra- and inter-institutional differences suggest it is not solely institutional wealth associated with graduate pay. We observe Canada is behind comparable countries in minimum funding levels for the next generation of scientists.

**Data availability statement:** The machine readable file and all code associated with the project are available from GitHub at https://github.com/UVicMicropaleo/Canadian-Minimum-Graduate-Stipends.

**Funding:** The author(s) received no specific funding for this work.

**Competing interests:** I have read the journal's policy and the authors of this manuscript have the following competing interests: Andrew Fraass is a tenure track professor who receives funding from Natural Sciences and Engineering Research Council of Canada (NSERC Canada, RGPIN-2022-03305 and DGECR-2022-00141) and has received funding from National Science Foundation, United States of America. Thomas Bailey is a graduate student who receives funding indirectly from NSERC. Thomas Bailey, Kayona Karunakumar, and Andrea Wishart are involved with Support Our Science, a not for profit grassroots advocacy group for graduate student and postdoctoral funding. They do not receive any financial compensation from this organization. This is discussed in the positionality statements of the paper. This does not alter our adherence to PLOS ONE policies on sharing data and materials.

## Introduction

Graduate students are an engine of scientific work for the academic system [1]. Despite this critical role in the Canadian research ecosystem, students report high levels of financial stress [2] exacerbated by increasing cost of living [3].

The following is a general description of the financial model of Canadian graduate studies in natural science: stipends provide financial assistance to students for living expenses (e.g., tuition, rent), as opposed to financial support for the costs associated with the research itself (e.g., reagents, equipment) [4]. Graduate students in Canada pay tuition and fees to their institution; waivers are rarely provided. When a stipend is provided, tuition and fees are paid by the student to their institution using their stipend, which itself was previously paid by the institution to the student. A master's degree typically takes 2 years to complete and a PhD 4–6 years [5,6]. Both master's and PhD students have a certain number of courses they must take, but these are often completed in the first few years allowing full time focus on research later in the degree.

Departments or equivalent academic units frequently set minimum stipend levels but individual principal investigators (PIs) can choose to exceed that amount. Stipends are typically funded from multiple sources: the institution (e.g., faculty of graduate studies provides the department funds based on number of students in their programs), a PI's research grants (see below), student awards, and/or teaching assistantships. TAships are frequently a core part of funding packages, with the pay earned from this job (which can be on the order of $3-6k/term) often included in the value of the minimum stipend. Typically, TAships are for 10 hours a week [e.g., 7] and reduce the capacity of the student to conduct research.

The Natural Sciences and Engineering Research Council of Canada (NSERC) Discovery Grant (DG) is a 5-year operating grant awarded to individual PIs and the foundation of most academic Canadian natural science graduate funding and research. Grant applications are judged evenly across three categories: a record of past research, highly qualified personnel (HQP; students, postdocs, and other mentees) produced by the researcher, and a proposed project [8]. Quality mentorship is therefore paramount in acquiring funding. Policy requires NSERC panels to not judge a researcher based on the number of HQP [8]; rather, applicants are judged on mentorship quality, post-mentorship careers, and so on. However, an incentive still remains to have more students, as publications and citations correlate positively with the number of HQP [9]. This pressure to train a large number of students with the relatively small operating grants may contribute to the suppression of stipend minimum support.

As a case study, we examined whether the guaranteed minimum stipends for graduate students in Physics and Ecological Sciences/Biology are a) enough to live on, b) comparable between programs and fields, c) related to the financial size of universities, and d) similar with those in two peer nations (US and UK).

## Positionality

Given the impact of career stage and nationality on perspectives about graduate programs, we summarize our positionality thusly:

*Bailey* is a British PhD student (Physics) at the University of Ottawa (Canada). He currently receives stipend support though a NSERC grant awarded to his supervisor. He is a department steward for his TA union and is on the Executive Council of Support Our Science, a grassroots organization advocating for increased pay for graduate students and post-doctoral fellows in Canada.

*Fraass* is an American tenure track Assistant Professor (Micropaleontology) at the University of Victoria (Canada). His graduate degrees and two postdoctoral positions were US based, followed by three years in England as a postdoc and fellow. He currently receives funding from NSERC and mentors graduate students.

*Karunakumar* is a Canadian early-career science policy professional. Her undergraduate and graduate education includes training in public policy and government relations. She is pursuing a career in science policy. Additionally, she serves as a volunteer for Evidence for Democracy (a fact-driven, non-partisan, not-for-profit organization advocating for the transparent use of evidence in government decision-making in Canada).

*Wishart* is a Canadian early-career academic publishing professional currently employed by Canadian Science Publishing. Her degrees (BSc, MSc, PhD) in biology were at Canadian universities, save for one year abroad (UK). She previously held institutional scholarships and received stipend support through operating grants (e.g., NSERC DG) awarded to her supervisors. In addition to leadership roles advocating for graduate students and performing research on graduate student experiences, she co-founded Support Our Science (a grassroots organization advocating for increased pay for graduate students and postdoctoral fellows in Canada) in 2022 and currently serves on its Board of Directors.

## Data sources

Publicly available information regarding stipends and waivers for the 38 universities in Canada offering graduate studies in Physics and/or Ecological Sciences/Biology (or the most similar department) was collected between February 16 to August 16, 2024. This study is limited to these two NSERC-funded programs to simplify the lengthy and difficult data collection process (see Accessibility), and because data are only valid for a single year. Tuition, mandatory fees, and minimum stipend amounts for each degree program were collected (see Supplementary Material for full details). When not discoverable on department websites or in associated files (e.g., graduate student handbooks), information was requested from program contacts via email.

## Accessibility

Determining accurate annual stipends was difficult, despite two of four authors possessing PhDs. It was rare for tuition and fees to be displayed in logical and easily interpretable ways. For example, most institutions split fees into different categories (e.g., bus pass, graduate association fees), but they commonly use different units of time to display each: some fees appeared per term, some per year, some per credit, while others had different values during fall, winter, and summer or for part-time vs. full-time students, and so on. This complexity made it extremely easy to make simple errors due to the number of parameters and unclear language/presentation. Many institutions include certain fees (e.g., healthcare) on their fee lists, others do not. Furthermore, several institutions put dollar amounts behind several menus and/or 'opt-out/opt-in' paperwork, or even private intranet pages.

Whatever the factors contributing to low discoverability and transparency in tuition and stipend data may be, the impact is obfuscated financial information prior to being enrolled in a program. One solution for these issues would be to have a "Finances" page clearly indicated on department websites, with a table laying out common funding scenarios, minimum stipends, durations, costs, and how much of the stipend is left after fees and tuition. A department that models this well is the University of British Columbia Physics Department website (https://web.archive.org/web/20240501083537/ https://phas.ubc.ca/graduate-program-financial-support).

## Stipend amounts

We collected tuition, fee, and stipend data from public-facing program and/or university websites for both domestic and international MSc and PhD students. We focus our main analysis on domestic students due to the complexity and

uncertainty around much of the international student data that was available. We include international data in the Supplementary Online Material for reference, but acknowledge the lower confidence in the provided values.

Data analysis was performed in R (v. 4.4.0 [10]). All code and data are publicly archived on GitHub (https://github.com/UVicMicropaleo/Canadian-Minimum-Graduate-Stipends). We defined Gross Minimum Stipend (GMS) as the minimum annual stipend, Net Minimum Stipend (NMS) as the minimum annual stipend after both tuition and fees are repaid to the institution, and MBM Shortfall as NMS minus the inflation-adjusted Market Basket Measure poverty threshold (MBM), for a single individual with no dependents assessed by Statistics Canada [11]. The dataset contains 140 programs from 38 institutions. We found 91 programs providing a guaranteed minimum stipend, while 21 programs provide no minimum stipend. We could not identify minimums and received no response to email requests from 24 programs. Only considering departments which guarantee support, the mean domestic GMS in Canada is Can$23,933, and mean NMS is Can$16,528 (Fig 1). An average program with a minimum stipend charges Can$7,585 in tuition and fees to domestic graduate students, with a mean ~33% (range: 18% - 61%) of the GMS repaid to their institution.

We compared stipend values to both governmental (MBM threshold [8], Fig 1) and private (a national report of rents for September 2023 by Rentals.ca [13] both with and without adding $12k/yr for food and incidentals) metrics (Supplemental Information). Only two programs (University of Toronto Physics PhD and MSc) appear to have domestic net minimum stipends which reached MBM thresholds, though these were estimated as fees were hidden behind an intranet page. The average department would need to raise their guaranteed minimum stipend by ~Can$9,584 for domestic students and ~Can$16,953 for international students to reach the MBM threshold.

NMS best demonstrates the disparity between Canadian, US [14–16], and UK stipends for PhD students (Fig 1) as tuition waivers are normally included for US or UK graduate studies. Notably, US data [12] (n = 215) is considerable, but not exhaustive, while the UK funding agency sets a single national stipend level. It is difficult to reasonably compare various indices of poverty in an unbiased way across national borders. Poverty metrics generated by different governments vary due to numerous factors (e.g., national and regional differences, political impacts of declaring a 'line of poverty'). Further, the non-TA portion of Canadian and UK stipends are untaxed while US stipends are taxed. US stipend data from [12] do not include ancillary fees, though it is extremely unlikely that fees or taxes are enough to erase the Can$23,759/yr

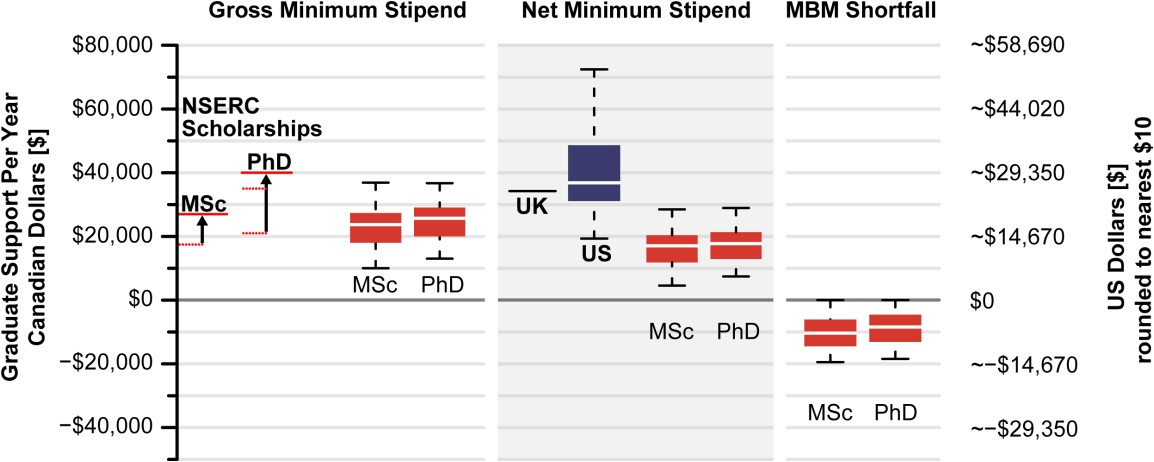

**Fig 1. Annual financial support in Canadian and US dollars to graduate students (MSc and PhD in physics and biology).** Red lines correspond to recently increased federal NSERC scholarships. Gross Minimum Stipend (GMS) is the guaranteed minimum funding for an MSc or PhD student provided by the institution. Net Minimum Stipend (NMS) is the GMS minus tuition and fees for an institution. MBM Shortfall is the NMS minus the Market Basket Measure (MBM), a poverty threshold, for an institution's location. US Biology stipends are from [12] and should be considered a rough approximation of the US stipends, without accounting for fees or taxes (see text).

difference between the means of the two countries (eg. the federal tax on a US$37k stipend for a single person would be around US$2,500). This difference is essentially the same as the mean GMS, suggesting in order to compete with the US or UK, observed Canadian stipends would need to double, compared to only increasing by ~1.5 times to meet the cost of living.

To identify if net stipend is driven by available institutional funding, we compared NMS against each university's endowment (n = 26 universities with n = 80 programs; Fig 2). We fit a generalized linear mixed model (package lme4 v.1.1.26, [17]) with university endowment (log10 transformed due to skewness of raw data), program (MSc or PhD), and field (Biology or Physics) as fixed effects, and university nested within province as a random effect because postsecondary funding falls within provincial jurisdiction. Endowment accounts for significant variation in NMS (S2 Table), suggesting a larger endowment generally results in higher stipends. We see similar effects when considering total expenses rather than endowment, suggesting that this is an effect predominantly of financial size of the institution (S3 Table) rather than specific use of endowments (although these parameters are highly correlated with larger institutions generally having larger endowments). Yet, substantial residual variation remains, even with accounting for variation across program and field within provinces (e.g., Ontario and Quebec demonstrate inter-institutional variation). Variation in stipends may result from a number of processes; e.g., different levels of support allocated to departments from the institution, restrictions on university endowment funds, different departmental budget priorities, or perhaps lagged responses to the increased expenses faced by students.

All analysis here is based on minimum stipend levels. Many graduate students do receive stipends higher than these levels, e.g. via top-ups from PI grants, additional TA hours, or external scholarships such as NSERC doctoral awards. However, the distribution of actual stipend values awarded is not publicly available and would require departments to report all stipend values, rather than simply department policies around guaranteed minimums (when they exist). Nevertheless, like minimum wage, minimum stipends are important to consider because they represent a lower bound that at least some graduate students experience. The purpose of setting a minimum is to ensure that all students receive

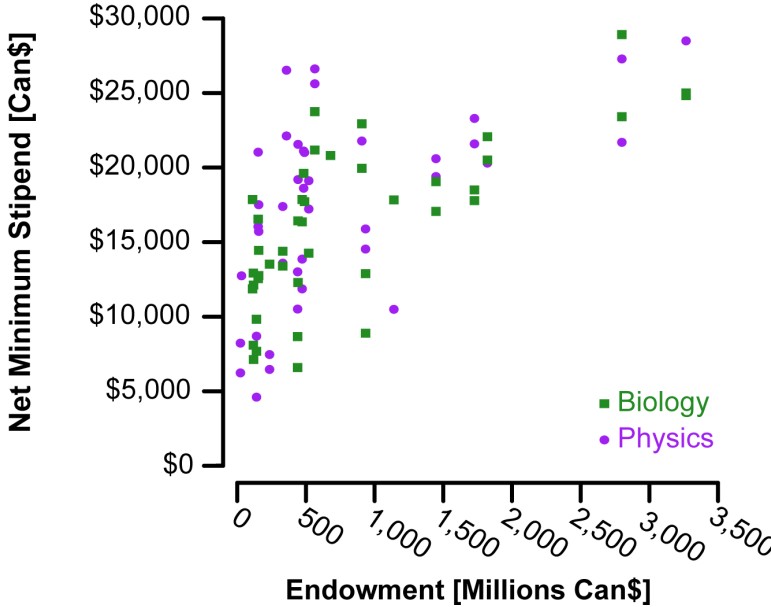

**Fig 2. Net minimum stipends in Canadian dollars to MSc and PhD graduate students in biology (green squares) and physics (purple circles) in relation to their institution's endowment in millions of Canadian dollars.**

financial support to enable them to focus on their studies without needing to seek employment elsewhere. If this is no longer being satisfied then the minimum level is not meeting the defined goals. Our mean/median minimum stipend levels are very similar to results from a recent survey [2], suggesting many students are, or are close to, receiving their departments minimum stipend.

There have been significant advances for Canadian graduate student finances recently. Federally sponsored scholarships received a large increase in 2024 [18], giving all PhD student recipients positive net stipends after accounting for tuition and MBM thresholds. However, for the vast majority of graduate students in Canada who are financially supported through other means, minimum departmental stipends are still essential to ensure financial viability of graduate program enrolment. For example, in 2021–22 only ~15% of domestic PhD students held a federal award. Further the proportion is likely to be lower for master's students and near zero for international students who are ineligible for the majority of awards. Whilst these proportions will increase as more scholarships are awarded following Budget 2024, it will still remain the minority of students.

## Conclusions

The complexity and opaque nature of the Canadian graduate funding system is likely to be unclear to potential graduate students evaluating program options. This is exacerbated by a lack of transparency around stipends and tuition at institutional and departmental levels, making comparison between departments difficult, along with other potential consequences (for example possibly posing a further barrier to recruiting underrepresented students [19]). It also, given the dissimilarity with other similar nations, makes it incredibly challenging for international students to make informed decisions due to the overly-complex nature of Canadian tuition and fee structures. While individual departments are unable to transform this system, they can improve their own transparency to allow for more informed choices by future graduate students.

Canadian and foreign students receiving the observed minimum stipend levels are poised to incur substantial debt to undertake graduate education, unless otherwise wealthy. This presents a series of problems; for example: equity, incurred health costs of poverty on the next generation of scientists, and the breadth of scientific inquiry [20]. Students would benefit from significant changes at whatever level is possible [21]. Governments [22]) have the ability to tweak aspects or alter the entire national system at once. However, within the current Canadian system, institutions, departments, or individual PIs can adjust minimum stipend levels. There is ample anecdotal evidence that this is already occurring at least at the departmental level, as several departments had increased stipends while we were performing quality control checks on our data. That minimum stipends are still low, however, suggests that individual departments probably cannot do this on their own, and may require assistance from institutions and/or government.

It is clear that Canada is behind competing countries when it comes to funding the next generation of scientists. Canadians who desire higher STEM education have three options: hope for significantly higher guaranteed support from a supervisor, department, or awards; incur substantial debts; or emigrate.

Graduate students, both in Canada and around the world, help drive academic advances in science [1]. Providing adequate financial support to these researchers enables this work to be done most effectively and boosts the ability of the entire scientific community to make progress. Not providing adequate support is damaging [23]. How best to provide this support is a global challenge, with countries offering a range of different systems and amounts. Regardless of the financial model offered, there is a potential for participation in science and innovation to be reduced when junior scientists are expected to live below the poverty line.

## Supporting information

**S1 data. Supplemental information file includes a short description of additional information gathered during data collection and rubrics for scoring data transparency.**
(DOCX)

**S1 Table. Mean scores (± standard deviation) for discoverability and transparency of tuition and stipend data.** Tuition transparency was assessed as ease of parsing, while stipend transparency was assessed as completeness of presented data. Scores were assigned based on the rubrics presented in Supplementary Online Material. Higher values correspond to greater discoverability and transparency, while lower values correspond to lower levels of both.
(DOCX)

**Supplemental Fig 1. Supported domestic minimum stipends compared to Rentals.ca values. Boxes at left are Net Minimum Stipends minus a local assessment of rents from Rentals.ca [13], while boxes at left subtract an additional $12k to account for food and other costs.**
(DOCX)

**Supplemental Fig 2. Supported domestic minimum stipends divided by province. Gross Minimum Stipend (GMS) is the guaranteed funding for a student provided by the institution. Net Minimum Stipend (NMS) is the GMS minus tuition and fees for an institution. MBM Shortfall is the NMS minus the Market Basket Measure (MBM) for an institution's location.**
(DOCX)

**Supplemental Fig 3. Annual financial support in Canadian and US dollars to international graduate students (MSc and PhD in physics and biology). Gross Minimum Stipend (GMS) is the guaranteed minimum funding for an MSc or PhD student provided by the institution. Net Minimum Stipend (NMS) is the GMS minus tuition and fees for an institution. MBM Shortfall is the NMS minus the Market Basket Measure (MBM), a poverty threshold, for an institution's location. US Biology stipends are from [12] and should be considered a rough approximation of the US stipends, without accounting for fees or taxes (see text).**
(DOCX)

**S2 Table. Summary of generalized linear mixed model fit for Net Minimum Stipend (NMS) as a function of university endowment using package lme4 v.** 1.1.26[17]. SEM = Standard error of the mean; SD = Standard deviation. Asterisk * denotes significance at α = 0.01.
(DOCX)

**S3 Table. Summary of generalized linear mixed model fit for Net Minimum Stipend (NMS) as a function of university expenses using package lme4 v.** 1.1.26 [17]. SEM = Standard error of the mean; SD = Standard deviation. Asterisk * denotes significance at α = 0.01.
(DOCX)

## Acknowledgments

M. Gaynor is thanked for their help with the US data. M. Gaynor, S. Laframboise, and K. Kharas are thanked for a preliminary review. Two reviewers (Lisa Walsh and an anonymous reviewer) are thanked for their careful consideration and comments.

## Author contributions

**Conceptualization:** Thomas J. Bailey, Andrea E. Wishart.

**Data curation:** Thomas J. Bailey, Andrea E. Wishart.

**Formal analysis:** Andrew J. Fraass, Thomas J. Bailey, Andrea E. Wishart.

**Investigation:** Andrew J. Fraass, Thomas J. Bailey, Kayona Karunakumar, Andrea E. Wishart.

**Methodology:** Andrew J. Fraass, Thomas J. Bailey, Kayona Karunakumar, Andrea E. Wishart.

**Project administration:** Thomas J. Bailey.

**Resources:** Andrew J. Fraass.

**Software:** Andrew J. Fraass.

**Validation:** Andrew J. Fraass, Thomas J. Bailey, Andrea E. Wishart.

**Visualization:** Andrew J. Fraass, Thomas J. Bailey, Andrea E. Wishart.

**Writing – original draft:** Andrew J. Fraass, Thomas J. Bailey, Kayona Karunakumar, Andrea E. Wishart.

**Writing – review & editing:** Andrew J. Fraass, Thomas J. Bailey, Kayona Karunakumar, Andrea E. Wishart.

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
