## [Decision Letter · Decision Letter 0]

11 Dec 2024

PONE-D-24-49904Canadian science graduate stipends lie below the poverty linePLOS ONE

Dear Dr. Fraass,

Thank you for submitting your manuscript to PLOS ONE. After careful consideration, we feel that it has merit but does not fully meet PLOS ONE’s publication criteria as it currently stands. Therefore, we invite you to submit a revised version of the manuscript that addresses the points raised during the review process.

Please revise your manuscript carefully to address all points raised by R1 and R2. Ensure you address concerns about the rigor of your analysis, particularly the balance between advocacy and research. Specifically, R1 highlights the need for more detailed analysis and clarification of regional differences and their interpretations.

We look forward to receiving your revised manuscript.

Kind regards,

Ashlyn Swift-Gallant

Academic Editor

PLOS ONE

Journal Requirements:

2. You indicated that ethical approval was not necessary for your study. We understand that the framework for ethical oversight requirements for studies of this type may differ depending on the setting and we would appreciate some further clarification regarding your research. Could you please provide further details on why your study is exempt from the need for approval and confirmation from your institutional review board or research ethics committee (e.g., in the form of a letter or email correspondence) that ethics review was not necessary for this study? Please include a copy of the correspondence as an ""Other"" file.

3. Thank you for stating the following in the Competing Interests section: [I have read the journal's policy and the authors of this manuscript have the following competing interests:

Andrew Fraass is a tenure track professor who receives funding from NSERC (Canada) and has recieved funding from NSF (US).

Thomas Bailey is a graduate student who receives funding indirectly from NSERC.

Thomas Bailey, Kayona Karunakumar and Andrea Wishart are involved with Support Our Science, a not for profit grassroots advocacy group for graduate student and postdoctoral funding. They do not receive any financial compensation from this organization. This is discussed in the positionality statements of the paper.]. Please confirm that this does not alter your adherence to all PLOS ONE policies on sharing data and materials, by including the following statement: "This does not alter our adherence to PLOS ONE policies on sharing data and materials.” (as detailed online in our guide for authors http://journals.plos.org/plosone/s/competing-interests). If there are restrictions on sharing of data and/or materials, please state these. Please note that we cannot proceed with consideration of your article until this information has been declared. Please include your updated Competing Interests statement in your cover letter; we will change the online submission form on your behalf.

5. Please include captions for your Supporting Information files at the end of your manuscript, and update any in-text citations to match accordingly. Please see our Supporting Information guidelines for more information: http://journals.plos.org/plosone/s/supporting-information .

Reviewers' comments:

Reviewer's Responses to Questions

**Comments to the Author**

1. Is the manuscript technically sound, and do the data support the conclusions?

Reviewer #1: Partly

Reviewer #2: Yes

2. Has the statistical analysis been performed appropriately and rigorously? 

Reviewer #1: N/A

Reviewer #2: Yes

3. Have the authors made all data underlying the findings in their manuscript fully available?

Reviewer #1: Yes

Reviewer #2: Yes

4. Is the manuscript presented in an intelligible fashion and written in standard English?

Reviewer #1: Yes

Reviewer #2: Yes

5. Review Comments to the Author

Reviewer #1: PONE D-24-49-49904

It is difficult to discern the difference between advocacy and research in this paper. I do not know what PLOS ONE’s editorial position is regarding the difference. If advocacy is within the journal’s mandate, with revision the paper perhaps warrants publication. As research it does not.

The most constructive way to explain this is to say that, like other papers of this kind it would toward the end of section on “limitations.” There are several.

1 Sponsored research is funded so differently in the U.S. and U.K. that the validity of evidence from those jurisdictions and comparisons to them are questionable. Perhaps less so for Australia. This makes me think about the findings of studies about policy diffusion and policy dissonance. There is a lot of unacknowledged dissonance here.

2 I appreciate the difficulty of getting data from only two departments, only minimum stipend values, and only NSERC, but as a matter of fact the simplification that the study admits fails to support most of the study’s conclusions. It makes the research a case study. Case studies, by definition, have little capability for generalization.

3 At the beginning the paper makes this assertion: Graduate students are a primary engine of scientific work for the academic system. And at the end we find this assertion as if it is a finding of the study: Graduate students, both in Canada and around the world, are the engine on which academic science runs.These are remarkable claims that, in my view, are more advocacy than research. As I read the Larivière paper it does not support the primacy that the paper gives to graduate student research in the first instance and the study itself provides no evidence in the second instance.

4 The paper’s first conclusion is about the “hidden curriculum” of an admissions process that results in under-representation in the academy. That may or may not be true, but it is an entirely different research question from where the paper begins, and it is hard to see how the study’s research methodology (which the paper presents very well) leads to that conclusion.

5 The discussion of proportionality is something like the “hidden curriculum” conclusion. The evidentiary basis of the conclusion is not clear, but more to the point it begs a question of an alternative that the paper does not admit: that the problem may be as much or more the number of stipends than the size of the stipends.

6 The study is correct to recognize the differences in stipend tax status between the U.S. and Canada (and maybe the UK, for which the study evidently did not collect comparable information. FYI Like Canada, stipends are not taxed in the UK.) This means that for comparative purposes the net minimum stipend in Canada is higher than the minimum stipend per se. This does not mean that Canadian stipends are inadequate and possibly still below the poverty line. It means that the comparative disparity may not be as great as presumed.

7 At several points the study refers to “substantial student debt” carried by graduate students. There are two potentially serious oversights here. First, approximately half of undergraduates in Canada graduate without debt. This varies from province to province. Second, the second oversight is the Canadian Education Tax Credit, which is available to graduates with or without debt.

8 Figure 2 is not conclusive. To be so would require a separation of restricted and unrestricted funds. Restricted means that a donor made a gift for a purpose specified – hence restricted – by the donor. Universities must comply with restrictions as long as they are not illegal. Of course, universities can refuse to accept restrictions. So, it is only the unrestricted portion of an endowment that can be directed to stipends. The study perhaps should come to different conclusion: that research-intensive universities in their fund-raising should place a higher priority on stipends.

Reviewer #2: This article highlights an important issue facing graduate students and the academy in Canada. For readers less familiar with the graduate systems in Canada and the US, the manuscript would benefit from additional details. I've provided some questions in the PDF to help you fill in some of the missing information.

Some attention to grammar and consistency within a sentence would further improve the manuscript.

6. PLOS authors have the option to publish the peer review history of their article (what does this mean? ). If published, this will include your full peer review and any attached files.

**Do you want your identity to be public for this peer review?** For information about this choice, including consent withdrawal, please see our Privacy Policy .

Reviewer #1: No

Reviewer #2: **Yes: ** Lisa Walsh

---

## [Author Response · Author response to Decision Letter 1]

11 Feb 2025

A text with formatting distinguishing between reviewer comments and our responses has been added to the files on this manuscript.

We greatly appreciate the reviews of our manuscript. We have done our best to address all of the reviewers' points within the manuscript itself. We have made minor changes to the text of the manuscript to clarify issues noted by the reviewers and have added a supplemental figure.

In particular, reviewer 1 suggested that we had been engaging in activism rather than research. We have done our best to be as clear as possible about the line between advocacy and academic research, and, if it deemed necessary by the editors, would be happy to discuss that further. We were keenly aware during all stages of this project that we should be aware of the line, and did our best to make it clear that we are not entirely impartial (e.g., positionality statements - and we would argue that no one in academia can be ‘impartial’ in this discussion). Further, we have done our utmost to only rest on the facts of the matter, rather than taking policy stances, for example. If we have strayed over the line, we would again welcome a discussion of where we can pull our toes back.

Thank you,

Andrew Fraass,

Reviewer 1

PONE D-24-49-49904

It is difficult to discern the difference between advocacy and research in this paper. I do not know what PLOS ONE’s editorial position is regarding the difference. If advocacy is within the journal’s mandate, with revision the paper perhaps warrants publication. As research it does not.

We have made a number of changes to avoid advocating for any specific policy outcomes and to remain firmly grounded in research and science.

In the abstract (Line 48-50, Line numbers correspond to the Track Changes version of the manuscript) we removed the sentence outlining potential consequences of of low minimum stipend levels - instead ending the the abstract with our results on Canada’s relative funding levels of graduate students.

In line 87, we qualify our statement that incentives to have more students contributes to the suppression of minimum stipends.

In line 245-246, we revised our comment following our discussion that stipends not large enough to enable students to focus on their studies “must be adjusted” to “not meeting the defined goals”, making it clear that our research finds that minimum stipends are failing to meet their intended purpose, rather than suggesting that our research prescribes a policy outcome.

In line 272, we removed language about departments ‘serving as agents for change’ and instead focus on an option that departments have to improve transparency.

In line 282-284, we removed policy suggestions such as standard stipend or tieing stipends to local cost of living and instead focus on the broad power of the government to modify the system.

In line 288-290, we qualify our assessment that the current level of minimum stipends is due to internal limitations in institutions and departments.

In line 295-296, we remove the suggestion that Canadian policy must change to attract and retain competitive talent. Whilst likely true, this is not a consequence of this research and so has been removed from this paper.

In line 303-309, we remove reference to ‘requiring’ anything and focus just on the potential consequences rather than action that would be needed to address these.

The most constructive way to explain this is to say that, like other papers of this kind it would toward the end of section on “limitations.” There are several.

Sponsored research is funded so differently in the U.S. and U.K. that the validity of evidence from those jurisdictions and comparisons to them are questionable. Perhaps less so for Australia. This makes me think about the findings of studies about policy diffusion and policy dissonance. There is a lot of unacknowledged dissonance here.

We acknowledge science is funded in significantly different ways in different jurisdictions and in no way are implying any challenge to that statement. However, we are trying to examine the difference in the size of funding received by graduate students, an outcome of science funding systems which arises regardless of the specific mechanisms used and is directly relevant to Canada’s international competitiveness and the welfare of graduate students performing research. Additional countries (and funding models) would have strengthened the comparison, but we were limited by capacity and chose the countries that the authors are most familiar with. In only one case (where we suggested the options of “enforcing a standard stipend level for all students”) did we propose that an aspect of another country’s funding system could be applicable in Canada (and which has now been removed) - otherwise the focus was on comparing outcomes in terms of stipend size for the different systems.

I appreciate the difficulty of getting data from only two departments, only minimum stipend values, and only NSERC, but as a matter of fact the simplification that the study admits fails to support most of the study’s conclusions. It makes the research a case study. Case studies, by definition, have little capability for generalization.

We acknowledge that as originally written the paper overstates the generalizability of the conclusions from a case study of these two departments. The following changes have been made:

Edited title to clarify the scope is the natural sciences.

Line 91: Clarified that this is a case study looking at physics and ecological sciences/biology. Correspondingly, we qualified claims made throughout the paper to remove an implication that they generalize to all other sciences.

We note, however, that roughly 33% of new NSERC doctoral scholarships and post doctoral fellowships in 2024 were awarded to individuals with the key words physics, biology, ecology, physique, biologie, or écologie in their reported department or discipline (with care taken to avoid double counting), suggesting that the case study of these two departments is probably applicable for around ⅓ of NSERC awards in and of themselves. https://www.nserc-crsng.gc.ca/NSERC-CRSNG/FundingDecisions-DecisionsFinancement/ScholarshipsAndFellowships-ConcoursDeBourses/index_eng.asp?year=2024

Further, we chose these two fields because of their historical reputations for being large (sample size) and well funded, but fairly ‘generic’ funding structures (ie., engineering can have relatively lucrative funding ties to industries). Humanities, it should be uncontroversial to say, is not funded at the level of STEM fields, and thus whatever the findings of this research might be, the conditions won’t be applicable there. Given the similarity between Physics and Biology, it seems that doing a similar survey of chemistry, math, or Earth sciences would not result in a drastically different result.

At the beginning the paper makes this assertion: Graduate students are a primary engine of scientific work for the academic system. And at the end we find this assertion as if it is a finding of the study: Graduate students, both in Canada and around the world, are the engine on which academic science runs.These are remarkable claims that, in my view, are more advocacy than research. As I read the Larivière paper it does not support the primacy that the paper gives to graduate student research in the first instance and the study itself provides no evidence in the second instance.

We have rephrased those statements in order to make it weaker. They now read “Graduate students are an engine of scientific work for the academic system.” (52-53) and “Graduate students, both in Canada and around the world, are an engine on which academic science runs.” (298-299) We do acknowledge that those might have been overstatements depending on a number of factors (fields, etc), and hope that pointing out that graduate students produce a substantial portion of research within the system is uncontroversial.

The paper’s first conclusion is about the “hidden curriculum” of an admissions process that results in under-representation in the academy. That may or may not be true, but it is an entirely different research question from where the paper begins, and it is hard to see how the study’s research methodology (which the paper presents very well) leads to that conclusion.

We acknowledge that introducing the concept of the ‘hidden curriculum’ in the conclusion was probably not the appropriate location. The aim of this paragraph was to highlight the challenge to potential new graduate students of understanding and finding information about graduate student stipends and tuition (with an example of the harm that this could potentially cause). We have reframed this paragraph by removing reference to the ‘hidden curriculum’ and making it clear that the focus is on the lack of transparency.

The discussion of proportionality is something like the “hidden curriculum” conclusion. The evidentiary basis of the conclusion is not clear, but more to the point it begs a question of an alternative that the paper does not admit: that the problem may be as much or more the number of stipends than the size of the stipends.

It is somewhat unclear what ‘discussion of proportionality’ this comment is in reference to. It is also unclear to us what the reviewer means by ‘the problem of the number of stipends’. Is this the scenario where, by reducing the number of graduate students, a department is able to raise its minimum stipend level without increases in external funding, or does it refer to the scenario in which the number of graduate students in departments which do not currently have any minimum stipend level being a greater problem? While we acknowledge that the latter is concerning, we base our conclusions only on data for those departments which have already accepted the principle of a minimum stipend. Further, it seems that suggesting that the number of graduate students be lowered in Canada in order to raise the minimum would be advocating for a specific policy? We did our best to point out the small size of the minimum stipends in Canada and not advocate for a specific solution, like lowering the number of graduate students or increasing the base NSERC funding to PIs. We agree however, that could be a factor, though in order to address that hypothesis we would need historical data.

The study is correct to recognize the differences in stipend tax status between the U.S. and Canada (and maybe the UK, for which the study evidently did not collect comparable information. FYI Like Canada, stipends are not taxed in the UK.) This means that for comparative purposes the net minimum stipend in Canada is higher than the minimum stipend per se. This does not mean that Canadian stipends are inadequate and possibly still below the poverty line. It means that the comparative disparity may not be as great as presumed.

In response to the other reviewer’s comments, we have included an example for the amount of federal tax a typical student may pay ($2500 on $37k stipend). We feel this sufficiently clarifies the magnitude of the disparity that is likely to be reduced by tax and strengthens our statement that it is “extremely unlikely that fees or taxes are enough to erase the Can$23,666/yr difference between the means of the two countries”. We have also included information that UK stipends are untaxed.

At several points the study refers to “substantial student debt” carried by graduate students. There are two potentially serious oversights here. First, approximately half of undergraduates in Canada graduate without debt. This varies from province to province. Second, the second oversight is the Canadian Education Tax Credit, which is available to graduates with or without debt.

In all three cases we mentioned debt, it was always in the context of debt incurred, not carried. The purpose of these statements was to highlight the immediate imbalance of stipend income and necessary living expenses and in no way a comment on the costs of carrying debt into the future.

While the tuition tax credit can be sizable, in reality for most graduate students it is not meaningful whilst they are graduate students. Since the bulk of stipends are tax free, graduate students typically pay minimal tax during their course of study. This means that the non-refundable tuition tax credit is unable to boost their net income and will be carried forward to apply once they are earning more taxable income (and presumably no longer a graduate student).

We as authors feel that adding a discussion of either of these points would be confusing rather than clarifying to the main points of the article, and so have chosen not to include new discussion points in the manuscript. However, should an editor want it to be included, we can.

Figure 2 is not conclusive. To be so would require a separation of restricted and unrestricted funds. Restricted means that a donor made a gift for a purpose specified – hence restricted – by the donor. Universities must comply with restrictions as long as they are not illegal. Of course, universities can refuse to accept restrictions. So, it is only the unrestricted portion of an endowment that can be directed to stipends. The study perhaps should come to different conclusion: that research-intensive universities in their fund-raising should place a higher priority on stipends.

The effect that we see is more to do with institution size, and as a proxy perhaps for the number and types of supports it represents (e.g., larger libraries, access to technology and specialized equipment, opportunities to collaborate, etc.) rather than the financially available component of endowment specifically (although institution size and endowment are highly correlated). Therefore, we did not find discussions about the restricted nature of endowments to be directly relevant. This was not clear in our original manuscript, and so we have added a line (Line 225-228) to clarify that a similar effect is seen when considering total expenses and included an additional figure in the supporting material to this effect. We have also added (Line 232) an explicit mention that restrictions on university endowment funds may be a source of the variation.

Reviewer 2

Can you provide the average and/or median years students in the two disciplines evaluated take to graduate with a Master’s /PhD? Are they taking classes each year, or do late-stage PhD students exclusively focus on dissertation research?

For all Master’s students the average time to graduate is around 2.1 years and for PhD students 5.4 years; however, Statistics Canada has not released this information with a breakdown by field of study and may not be informative for the focal programs in our study. The number of classes required varies by programme and university, but typically are completed in the first few years of the degree allowing exclusive focus on research (and possibly TAing) for the latter part of the degree.

Is the stipend greater for TAships? Are they expected to do less research if teaching?

Typically the value of the stipend is set without explicit consideration of TAships, though they are often a requirement to receive the amount of stipend offered. In some cases it’s up to the supervisor to determine if TAing is required (e.g., whether they can pay an equivalent amount to the TA salary from their research grants). It is possible that some students (particularly if they hold an external scholarship) are able to TA and receive a larger total income, though these additional TA positions are not going to be guaranteed.

TAships in Canada are typically for 10 hours a week (although there is variation around this). Holding a TAship will reduce the capacity of a student to conduct research than if they were funded in an alternative manner.

It is notable that this is extremely different from a TA in the United States, in that doing a Canadian TAship is effectively sometimes a small additional payment on top of the stipend, and does not provide a tuition waiver or anything other than an additional amount of money. It’s a fairly different position with a smaller time commitment but also substantially fewer benefits in Canada. We have tried to make this

---

## [Editor Report · Decision Letter 1]

11 Mar 2025

Canadian natural science graduate stipends lie below the poverty line

PONE-D-24-49904R1

Dear Dr. Fraass,

We’re pleased to inform you that your manuscript has been judged scientifically suitable for publication and will be formally accepted for publication once it meets all outstanding technical requirements.

Kind regards,

Ashlyn Swift-Gallant

Academic Editor

PLOS ONE

Additional Editor Comments (optional):

One minor editorial request: Please provide sources in the introduction for the length of time it takes to complete a master’s and phd degree, as well as the typical TAships hours/week.
---

## [Editor Report · Acceptance letter]

PONE-D-24-49904R1

PLOS ONE

Dear Dr. Fraass,

I'm pleased to inform you that your manuscript has been deemed suitable for publication in PLOS ONE. Congratulations! Your manuscript is now being handed over to our production team.

Kind regards,

on behalf of

Dr. Ashlyn Swift-Gallant

Academic Editor

PLOS ONE